# Association of Physical Activity and/or Diet with Sleep Quality and Duration in Adolescents: A Scoping Review

**DOI:** 10.3390/nu16193345

**Published:** 2024-10-01

**Authors:** Jon Cruz, Iñaki Llodio, Aitor Iturricastillo, Javier Yanci, Silvia Sánchez-Díaz, Estibaliz Romaratezabala

**Affiliations:** 1Faculty of Education and Sport, University of the Basque Country, UPV/EHU, 01007 Vitoria-Gasteiz, Spain; joncruzarroniz@gmail.com; 2AKTIBOki: Research Group in Physical Activity, Physical Exercise and Sport, Physical Education and Sport Department, Faculty of Education and Sport, University of the Basque Country, UPV/EHU, 01007 Vitoria-Gasteiz, Spain; inaki.llodio@ehu.eus (I.L.); aitor.iturricastillo@ehu.eus (A.I.); javier.yanci@ehu.eus (J.Y.); estibaliz.romaratezabala@ehu.eus (E.R.); 3Society, Sports and Physical Exercise Research Group (GIKAFIT), Physical Education and Sport Department, Faculty of Education and Sport, University of the Basque Country, UPV/EHU, 01007 Vitoria-Gasteiz, Spain; 4Physical Activity, Exercise, and Health Group, Bioaraba Health Research Institute, 01007 Vitoria-Gasteiz, Spain; 5Department of Education and Humanities, Universidad Europea de Madrid, 28670 Madrid, Spain

**Keywords:** exercise, sport, diet, nutrition, sleep, insomnia

## Abstract

**Background:** Sleep is essential for health, especially during adolescence. However, most adolescents do not obtain the recommended 8 to 10 h of sleep, and their health is significantly affected. While both physical activity (PA) and diet have been shown to help improve the sleep quality and duration, the combined association of these two factors with sleep has yet to be analysed. **Objectives:** Therefore, the main objective of this study was to assess the evidence on the combined association of PA and diet with the quality and duration of sleep in adolescents. Secondary objectives were to analyse the evidence on the single association of PA with the quality and duration of sleep in adolescents and to analyse the single association of diet with the quality and duration of sleep in adolescents. **Methods:** To this end, a scoping review was conducted with a structured search in four online databases (PubMed, Scopus, Web of Science and ERIC). **Results:** The findings suggest that the amount of PA (time/week) and healthy dietary patterns, characterised by meal regularity and high consumption of fruits and vegetables, favour a better quality and a longer duration of sleep. Conversely, less weekly PA and the intake of less healthy foods, such as ultra-processed foods, are associated with decreasing the sleep quality and duration. **Conclusions:** In conclusion, the results underscore the importance of considering PA and diet as an appropriate approach to investigating sleep quality and duration in adolescents. Studies analysing the interplay between PA, diet and sleep in adolescents are scarce.

## 1. Introduction

Sleep, a fundamental biological need, is essential for health and well-being [1]. Recommendations for sleep duration vary depending on age. While young and adult individuals are recommended to sleep between 7 and 9 h per day [2], adolescents are advised to sleep between 8 and 10 h per day [3]. Despite these guidelines, international sleep duration statistics paint a worrying picture. Firstly, some studies indicate that most young adults fall within the lower range of sleep, sleeping only around 7 h [4]. Secondly, the situation is even more alarming among adolescents, with 11% to 30% experiencing sleep disorders. This variation in the prevalence of adolescents’ sleep disorders could be due to age, region and culture. For example, older adolescents go to bed later and sleep less [5]. In addition, Asian adolescents go to bed later than American adolescents and obtain less sleep than Europeans [6]. Similarly, Galan-Lopez et al. indicate that 68% of boys and 69% of girls aged 13 to 16 slept less than the recommended 8 to 10 h, and regarding sleep quality, 53% of boys and 44% of girls experienced a poor sleep quality [7].

It has been shown that, in addition to sleep, physical activity (PA) and diet are essential for health [8]. Diet, the process of selecting and consuming food [9], is a voluntary act and fundamental for health and well-being [10]. Therefore, it is essential to maintain an adequate and balanced diet [11]. Izquierdo et al. define a healthy diet as one that meets all nutritional needs at every stage of life, from childhood to old age [12]. As Calañas-Continente and Bellido report, a healthy diet should provide the necessary calories for metabolic and physical processes and an adequate supply of nutrients to help maintain an ideal weight, and reduce the risk of chronic diseases [13].

The World Health Organization (WHO) considers PA, diet (or healthy eating) and sleep as primary factors for good health and well-being [14,15,16,17,18]. Given the importance of PA, diet and sleep for adequate health and well-being, it may be interesting to probe their relationship. There is a substantial body of research on the positive association between PA and sleep. For example, Dolezal et al. highlighted that physical exercise and sleep have significant positive effects on each other, reinforcing the importance of both for overall health [19]. Additionally, Antczak et al. found that boys and girls who engage in vigorous PA enjoy better overall sleep, including a longer sleep duration [20]. De Souza et al. observed that adolescents who experienced low levels of PA and excessive screen time and who were overweight tended to have a shorter sleep duration [21]. This finding suggests that there may be a correlation between an active lifestyle and sleep duration. As a result, it may be reasonable to conclude that PA positively influences sleep [22], and similarly, that adopting healthy PA and sleep habits improves various health markers in school-aged children and young people [23]. However, the association looks complex and may vary by PA intensity; in this sense, moderate-to-vigorous PA (MVPA) is associated with an increased wake time per hour after sleep onset (TAPH) [24]. Additionally, research by Chaput et al. showed that a higher level of PA was associated with a shorter sleep duration and lower sleep quality [25]. Furthermore, Dolezal et al. showed that the association between PA and sleep can also vary by sex, with higher levels of PA associated with a lower prevalence of insomnia in boys [19]. It has been shown that the association between PA and sleep might be to a certain extent a mental affair. PA may cause increased psychological well-being; consequently, one may also be subjectively more pleased with one’s sleep [19].

Regarding the association between diet and sleep, Alibabaei et al. established a direct relationship between adherence to healthy diets and a longer sleep duration [26]. This link suggests that a healthy diet may be a crucial factor in improving adolescent sleep quality and duration. Additionally, research by Adelantado-Renau et al. showed a positive association between adherence to the Mediterranean diet and academic performance in adolescents, highlighting quality sleep as a central mediator in this relationship [27]. Likewise, Firouzi et al. observed that excessive carbohydrate consumption in children is associated with a shorter sleep duration and deteriorated sleep quality, which may contribute to a higher risk of overweight or obesity [28]. These results are also consistent with St-Onge et al. [29], who indicated that diet and the composition of different meals can have both chronic and acute impacts on sleep. In addition, it has been observed that sleep disturbances affect the type of food consumption; for example, sleep disturbance increases energy intake with a more pronounced desire to consume energy-dense foods [30]. In turn, a change in the type of macronutrients can affect different sleep parameters such as rapid eye movement (REM), slow-wave sleep (SWS) or waking after sleep onset (WASO), affecting to sleep efficiency [31,32].

Furthermore, some studies have pointed out that PA and diet together can modify sleep patterns during adolescence [33,34]. Specifically, Awad et al. found that physical exercise, total fat intake (negative correlation) and a low body mass index (BMI) were associated with increases in phase II sleep (REM sleep) [33], showing a complex and multifaceted interaction between PA, diet and sleep, and their influence on health and well-being. In turn, Mozaffari-Khosravi et al. also identified a direct relationship between low levels of PA [34], inadequate eating patterns and a poor sleep quality. Despite considerable advances in research on the relationship between PA, diet and sleep, there is one gap in the literature; in fact, while systematic reviews have so far been conducted to explore the association of PA with sleep [19,20,23,35] or the relationship between diet and sleep [26,36], no review has been found to comprehensively address the association of both variables with sleep in adolescents, that is, the effect of PA and diet together on sleep quality and quantity.

Therefore, the main objective of this scoping review was to assess the extant research on the combined association of PA and diet with the quality and duration of sleep in adolescents. Secondary objectives were to assess the evidence on the single association of PA with the quality and duration of sleep in adolescents and on the single association of diet with the quality and duration of sleep in adolescents.

## 2. Methods

This study was conducted following the Preferred Reporting Items for Systematic Reviews and Meta-Analyses (PRISMA) statement [37]. A structured search was performed in four online databases (PubMed, Scopus, Web of Science and ERIC) and concluded on 20 March 2023. The search terms included a combination of free-text words for key concepts related to PA, diet, adolescents and sleep. To enhance the search process, some Boolean operators (AND, OR and NOT) with field identifiers or qualifiers (Title, Title/Abstract, All) were used. The following search equation was written in the search boxes of the mentioned databases to find relevant articles: (adolescents OR “young people” OR teenagers OR youth OR “high school” OR “secondary school”) AND (“physical activity” OR exercise OR sports OR fitness) AND (“dietary habits” OR “eating habits” OR “food consumption” OR nutrition) AND (sleep OR “sleeping habits” OR “bedtime routines” OR rest) AND (“sleep quality” OR “sleep disturbance” OR insomnia).

The PECOS model was used to determine the inclusion criteria [38]: P (Population): adolescents (11 to 18 years), E (Exposure): daily PA and/or dietary habits, C (Comparators): levels of daily PA and/or dietary habits, O (Outcome): any measure of sleep duration or quality, and S (Study design): cohort studies, cross-sectional studies, clinical trials, observational studies and longitudinal studies.

No filters were applied to increase the analytical power, but articles were excluded if their title or abstract was not written in English or Spanish and the full text was in a different language than those mentioned.

### 2.1. Inclusion and Exclusion Criteria

Studies were included based on the following inclusion criteria: (1) sample consisting of adolescents aged 11 to 18 years without health alterations or pathologies; (2) studies evaluating PA and/or diet along with sleep; (3) studies with a quantitative research design (cohort studies, cross-sectional studies, observational studies and longitudinal studies) and (4) studies published in English or Spanish. Conversely, the following were excluded: (1) studies whose population was outside the 11 to 18 age range and did not disaggregate data for participants within this age range; (2) studies that, despite evaluating some of the mentioned variables, did not analyse the association of these variables (PA, diet or both) with sleep; (3) studies investigating the variables only during the lockdown period of the COVID-19 pandemic (these studies were excluded because during this period, the PA and sleep patterns changed considerably [39,40,41]); (4) review articles including systematic reviews and meta-analyses; (5) studies that were not available in the full text; (6) articles not focused on studying the effects of diet or PA on sleep; (7) articles published before the last ten years (before 2013) and (8) studies that conducted an intervention.

### 2.2. Included Studies

Articles were sought in databases, and citations of potentially relevant publications were selected and processed using the Rayyan program (https://www.rayyan.ai/, accessed on 22 March 2023), which helped to identify duplicates through comparison and cross-checking. If the citation showed potential relevance, the abstract was examined. All articles evaluated as eligible and classified as relevant (when abstracts indicated possible inclusion) were reviewed in full text. Any disagreement about whether the inclusion criteria were met was resolved through discussion between the first and last authors. The search for articles was also carried out using the snowball strategy [42], whereby the reference sections of all relevant articles were examined. Nevertheless, this strategy did not produce more articles for the scoping review. Finally, inclusion and exclusion criteria were used to select articles classified as relevant for inclusion in this review (Figure 1).

### 2.3. Data Synthesis

To obtain a more comprehensive view of the state-of-the-art research in this field and distinguish existing gaps, a narrative synthesis approach was used, which enabled an understanding of studies conducted to date in this research area and identification of existing gaps and research needs [43]. The identified studies were summarised in a table that including their main characteristics, comprising the first author and publication year, country of origin of the sample, study type, number of participants, age, sex, PA measurements, diet measurements, sleep measurements and results obtained from the analysis.

### 2.4. Quality Assessment

To evaluate the methodological quality of the studies, the Critical Appraisal for Cross-Sectional Research from the Joanna Briggs Institute (JBI) was applied [44]. This tool provides the ability to assess the methodological quality of a cross-sectional study and determine the extent to which bias has been excluded or minimised in its design, conduct and/or analysis. The JBI checklists classify the risk of bias into three categories. For analytical cross-sectional studies, a score of 0 to 3 indicates a high risk of bias, a score of 4 to 6 implies some concerns and a score of 7 to 8 indicates a low risk of bias.

Eight criteria were evaluated, including the adequacy of the sampling timeframe, recruitment procedure, sample size, description of participants and context, among others. Studies were assigned one point for each criterion met, and those with a score of five points or higher were included. Two authors individually assessed the quality of the articles and then discussed them together.

## 3. Results

### 3.1. Search Strategies and Included Studies

A total of 500 results were identified during the search through electronic databases. Of these results, 168 were identified on PubMed, 176 on Scopus, 151 on Web of Science and 5 on ERIC. After removal of duplicates (*n* = 147), 353 results remained. After a review of the titles and abstracts, 227 articles were excluded for not meeting the research objectives, leaving 126 full-text articles for a more detailed review, of which 113 were excluded for not meeting one or several of the established inclusion criteria. The reasons for excluding articles were as follows: studies conducted during the COVID-19 pandemic period (*n* = 36), no adolescents studied (*n* = 30), research objective and analysis different from the objective of the review (*n* = 28), intervention studies (*n* = 14) and review articles (*n* = 4). Some studies were excluded for multiple reasons. Figure 1 shows the results of the article search and selection process. In the end, a total of 13 studies met the inclusion criteria [24,34,45,46,47,48,49,50,51,52,53,54,55].

### 3.2. Studies Quality

The quality of the studies included in the review was carefully assessed using the Critical Appraisal for Cross-sectional Research checklist from the JBI, and the score of each study is presented in Table 1. With scores of 7 and 8, seven studies demonstrate a low risk of bias [24,45,49,50,52,54,55], while six studies raise some concerns [34,46,47,48,51,53] with scores of 5 and 6.

### 3.3. Study Characteristics

The studies included a total sample of 272,044 adolescents with mean ages ranging from 11.9 to 16.4 years (14.37 ± 1.21 years). The sample sizes of the studies ranged from 59 to 118,462 participants (M = 20,986.46; SD = 7392.13). The studies included in the review were conducted in Germany [24], Australia [48], Brazil [54], South Korea [49,51,52], the United States [46], Iran [34,45,50,55], Italy [48] and Turkey [53]. The studies were mainly cross-sectional in design (*n* = 12 studies), and there was one cohort study [24]. Of the identified studies, eight reported results related to diet and sleep [45,46,47,48,50,52,53,55], three studies reported results related to PA and sleep [45,49,51] and only two showed results related to diet, PA and sleep [34,51] (Table 2). However, most studies (*n* = 8) did measure diet, PA and sleep [34,45,47,48,50,51,54,55]. In addition, three studies measured diet and sleep [46,52,53], and two studies measured PA and sleep [24,49].

### 3.4. Measurement of Physical Activity (PA)

To measure PA, two studies used accelerometers [24,49], and one of them also utilised actigraphy [49]. However, most studies used questionnaires (*n* = 7 studies). Different types of questionnaires were employed: three studies used the Modifiable Activity Questionnaire (MAQ) [45,50,55], one study utilised the modified International Physical Activity Questionnaire (IPAQ) [54], one study used the Physical Activity Questionnaire for Adolescents (PAQ-A) [47] and three studies employed ad hoc questionnaires [34,48,51].

### 3.5. Dietary Measurement

All studies focusing on diet (*n* = 11) used questionnaires to monitor dietary intake and eating patterns, but the measurements and criteria employed varied widely. Five studies utilised a Food Frequency Questionnaire (FFQ) [48,50,51,52,55]; two studies assessed food intake over the past 24 h [48,53], one study employed a self-administered questionnaire analysing the frequency of consumption of snacks and ultra-processed foods [54] and, finally, three studies inquired about different aspects related to diet and eating patterns using questions of varying nature such as ‘How many times do you have breakfast in a week?’ or ‘Over the past seven days, how many times have you eaten fruit?’ [34,45,46].

### 3.6. Sleep Measurement

Among the articles that analysed sleep (*n* = 11), measurement was primarily conducted using questionnaires. However, two studies utilised accelerometers [24,49], and one also used actigraphy and a sleep diary [49]. Among the studies that used questionnaires, three employed the Insomnia Severity Index (ISI) to measure the severity of insomnia [45,50,55], two studies used the Pittsburgh Sleep Quality Index (PSQI) to assess sleep quality and duration [34,53], one study measured the frequency of insomnia over the past 12 months using a direct question [54], two studies asked about bedtime and wake-up time during the last five or seven days of the week [51,52], one focused on sleep patterns (bedtime, wake-up time, nap duration, etc.) [47] and two inquired about the number of hours of sleep; Bhurosy and Thiagarajah focused on the number of hours of sleep on a school day [46], while Golley et al. assessed the number of hours of sleep in the last 24 h on a school day and a weekend day [48].

## 4. Main Findings and Discussion

### 4.1. Relationship between Physical Activity and Sleep

Thirteen studies examined the association of diet and/or PA with sleep quality and/or duration [24,34,45,46,47,48,49,50,51,52,53,54,55]. Eight of these studies focused on the association of diet with sleep [45,46,47,48,50,52,53,55], three on the association of only PA on sleep [24,49,54] and two on the association of both diet and PA with sleep [34,51].

Several studies found that higher levels of PA were associated with a better sleep quality and duration [24,49,51,54]. According to Jindal et al., there is an association between less sedentary or lighter PA time and adequate sleep [49]. The study also found that more PA time was associated with an earlier bedtime, regardless of sleep duration. Similarly, Lee et al. observed a correlation between insufficient sleep (<7 h) and several risk factors, specifically more sedentary time and a lower likelihood of engaging in muscle-strengthening exercises [51]. Negele et al. found that PA improves sleep quality on the night following PA practice [24]. Although the mentioned studies show that PA and sleep are correlated, the specific association of PA with sleep can be complex and may vary by sex. For example, in girls, each 10 min increase in moderate-to-vigorous physical activity (MVPA) is associated with improved sleep efficiency (SE) [24], while in boys, performing at least 60 min of MVPA per day decreases sleep onset latency (SOL) [24]. For both sexes, MVPA was associated with increased SE [24]. In girls, MVPA was also associated with increased wake time per hour after sleep onset (TAPH), whereas in boys, MVPA was associated with lower SOL [24]. Werneck et al. provided another angle of investigation by examining PA and the prevalence of insomnia, as they observed that higher levels of PA were associated with a lower prevalence of insomnia in boys [54], a finding consistent with previous research [19]. In girls, in contrast, a non-linear relationship was found between PA and insomnia, as moderate levels of PA were associated with lower insomnia levels, while high levels of PA (≥420 min/week) were associated with higher insomnia levels [54]. The specific association of PA with sleep may also vary depending on timing. For some subjects, intense exercise too close to bedtime can disturb sleep, possibly through increases in circulating cortisol levels [54]. These findings highlight the importance of PA for sleep health, but they also indicate the need for further research to clarify the roles of timing, gender and PA intensity in these associations. In particular, further investigation into the inverse relationship between high levels of PA and insomnia in women would be beneficial.

PA and sleep can be measured subjectively and objectively with many instruments [56], and this diversity of assessment tools complicates a comparison of the results of different studies. Most studies analysed in the present review used subjective measures, i.e., questionnaires, which agrees with Lang et al., who found that most studies measuring PA and sleep use subjective methods [22]. It has been shown that subjective PA in adolescents is generally overestimated when compared with objective methods of measurement (actigraphy and heart rate monitoring). Regarding sleep, the correlations between sleep duration and sleep quality in adolescents are low or nonsignificant; as a result, they must be regarded as two separate domains. Whether a questionnaire focuses on quantitative or qualitative aspects of PA and sleep may affect measurement results [22].

### 4.2. Relationship between Diet and Sleep

Regarding the association of diet with sleep, several studies found that food choices and meal regularity may significantly influence sleep quality. On the one hand, Beigrezaei et al. [45], Bhurosy and Thiagarajah [46], Ferranti et al. [47] and Ozkan et al. [53] found that certain foods, such as spicy foods and sweets, were associated with a poorer sleep quality [45,47]. As for the association between food choices and sleep duration, Bhurosy and Thiagarajah, Ferranti et al., Golley et al. and Min et al. found that a longer sleep duration was associated with a diet rich in fruits, vegetables, milk and low-energy, nutrient-dense foods [46,47,48,52]. Similarly, consuming energy-dense, nutrient-poor foods, such as ultra-processed foods, was associated with shorter and poorer-quality sleep [50,51]. These results are consistent with findings from previous reviews, which concluded that a longer sleep duration may be associated with a healthier dietary pattern [26,57]. In this sense, Alibabaei et al. studied the relationship between dietary patterns and sleep duration and quality and showed that healthy dietary patterns (i.e., vegetables and greens, healthy vegetables and proteins, and fruits and vegetables) were associated with a prolonged sleep duration [26]. However, they found inconclusive results for the association between diet and sleep quality, where the results were more mixed and inconclusive as they could be affected by gender, PA, screen time, etc. [26]. It seems that meal timing and the bedtime also play an essential role in sleep quality. A poor sleep quality is related to late dinner and bedtimes [34]. Conversely, an early bedtime is associated with a better dietary quality and higher consumption of fruits and vegetables [48].

### 4.3. Relationship between Physical Activity, Diet and Sleep

Finally, two studies reported results related to PA, diet and sleep [34,54]. Lee et al. observed that less frequent strength training exercises (>3 times/week) and more frequent consumption of salty biscuits was associated with a shorter sleep duration [51]. Mozaffari-Khosravi et al. also found that both PA and dietary patterns influenced sleep quality; low levels of PA (<1 h/week), late dinner habits and a late bedtime were associated with a poor sleep quality. Furthermore, a better sleep quality was observed among Iranian students who had dinner before 22:00 than who had it after 22:00 [34]. Conversely, an early bedtime is associated with a better dietary quality and higher consumption of fruits and vegetables [48]. The association of PA and diet with sleep quality and duration in these studies was analysed in isolation, and neither of them analysed the possible additive effect of PA and diet, so further investigation may be necessary to disentangle the combined relationship of PA and diet on sleep. Despite these limitations, the studies analysed show that low levels of PA, insufficient strength training, unhealthy conditions and late eating habits (close to bedtime) lead to a shorter sleep duration. A shorter sleep duration has also been shown to be associated with going to bed late. To better understand the difference between the association of PA and diet with sleep quality and duration in isolation, and the possible additive effect of PA and diet when they are combined, multiple regression analysis could be used. This analysis would help elucidate if PA and diet together explain significantly more variation in sleep quality and duration compared to when analysed alone. In addition, to explore the relationship between the three variables, partial and semipartial correlations can also be used. These types of correlations are used to study the association of two variables of interest while statistically controlling other variables. Partial correlation is used to analyse the relationship of two variables of interest (for example, diet and sleep) while controlling the effect of other variables (PA) on the two variables of interest. Semi-partial correlations are used to analyse the relationship of two variables of interest (for example, diet and sleep) while controlling the effect of other variables (PA) on only one of the two variables of interest (on diet or on sleep, not on both). Studies using these types of correlations have been published in many fields including psychology, genomics, medicine and metabolomics [58]. Finally, the possible additive cause and effect relationship of PA and diet on sleep could be explored with direct interventional studies where, after a control period, participants could be randomly assigned to four different groups, and four different interventions could be performed (Group A diet 1 PA 1, group B diet 1 PA 2, group C diet 2 PA1 and group D diet 2 PA2), comparing within the groups and between the groups the quality and duration of sleep pre- and post- intervention.

## 5. Conclusions

Based on the current evidence, it seems that physical activity is associated with sleep, although the results outlined open up new research avenues of investigation, for instance, the sex-dependent relationship between the intensity of physical activity and insomnia. Furthermore, one clearer conclusion to be drawn from this review is that dietary patterns have a significant impact on sleep. Specifically, a balanced diet rich in fruits and vegetables and low in ultra-processed foods favours a better sleep quality and longer sleep duration. Similarly, consuming less healthy foods is associated with a decreased sleep quality and duration, and the timing of main meals and bedtime have also emerged as relevant factors. These findings are of particular interest for future interventions aimed at improving adolescents’ quality of life through physical activity and/or diet and sleep.

## 6. Study Limitations and Future Directions

Despite the high methodological rigour of this scoping review, this study has some limitations that should be considered when interpreting the results. This scoping review was restricted to articles published only in English or Spanish, which may have excluded relevant research published in other languages, potentially limiting the generalisability of our findings. This underscores the need for future research to use standardised studies to measure these variables and allow for more direct comparisons.

Furthermore, it is essential to note that the robustness of the conclusions is limited by the number of studies included in this review, totalling 13 studies, with only two reporting results related to the association of PA, diet and sleep. It is also worth highlighting that the studies examining these interactions did so partially, studying only pairs of variables and not the complete trio, which prevents a more comprehensive analysis of the combined association of PA and diet with sleep. Moreover, because in the present study, we were mainly interested in evaluating the evidence on the combined association of PA and diet with sleep quality and duration in adolescents, the nature of the studies analysed was correlative and they could not provide information on the cause–effect relationship between the mentioned variables.

In light of the limitations identified in this scoping review, it is evident that further research is needed to deepen our understanding of the association between PA, diet and sleep combined. Longitudinal and intervention studies should be implemented to explore the temporal and causal association between these variables in adolescents. Particularly, an emphasis should be placed on implementing study designs that evaluate the combined impact of PA and diet on sleep quality and duration, since studies examining these interrelationships are scarce and it is crucial to probe all of them together to better understand correlations in isolation and the possible additive effects when they are combined.

In addition, most measurements of PA, diet and sleep are based on self-reported questionnaires. In the future, it will be important to consider the use of more objective measures in order to avoid social desirability or memory biases. Moreover, although in the majority of the studies analysed, it was deduced or appeared that it was the adolescents themselves who completed the questionnaires on which the studies were based, this reality has not been proven or described clearly, since not all studies clearly indicate that this is the case, so it may be that in some cases, the questionnaires were completed in part or entirely by the parents of the adolescents. Therefore, in these latter cases, the links between PA, diet and sleep would be due more to parental decisions and not so much to those of the adolescents themselves. Despite this, these decisions would directly affect adolescents and, more importantly, the quantity and quality of their sleep. For several reasons, adolescents may spend a large amount of time away from home without parental control, and this causes them to lose control over their eating and PA habits during their leisure time. In this sense, direct intervention will be necessary both with the adolescents themselves and with their parents to make them aware of the detrimental effects of insufficient PA and an unbalanced diet on the amount and amount of sleep.

## Figures and Tables

**Figure 1 nutrients-16-03345-f001:**
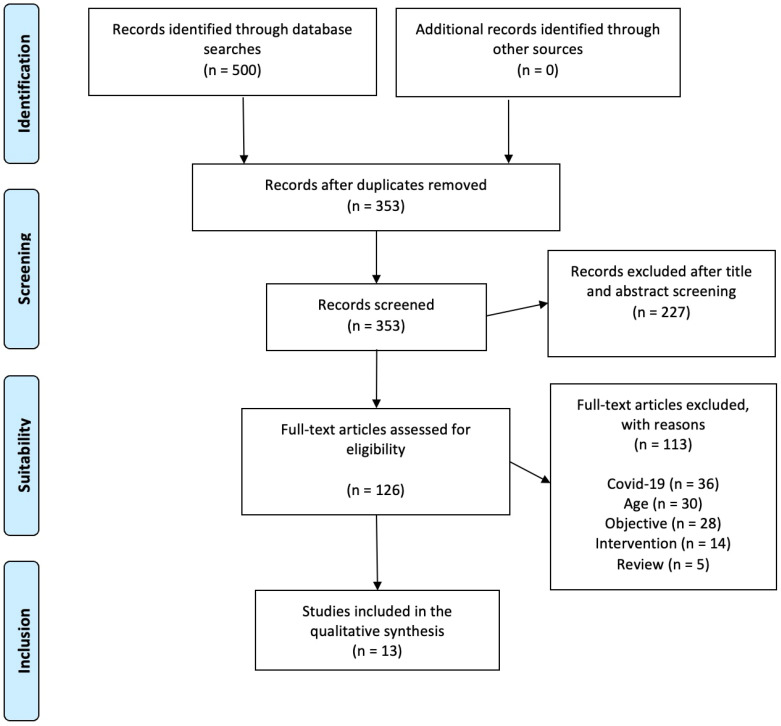
PRISMA flowchart for the identification, selection, eligibility and inclusion of studies.

**Table 1 nutrients-16-03345-t001:** Evaluation of studies using the Critical Appraisal for Cross-sectional Research Checklist from the Joanna Briggs Institute (JBI) for cross-sectional studies.

Reference	1	2	3	4	5	6	7	8	Score	Qualitative Assessment
Beigrezaei et al. (2022) [45]	1	1	1	1	1	1	1	1	8	Low risk
Bhurosy and Thiagarajah (2020) [46]	0	0	0	1	1	1	1	1	5	Some concerns
Ferranti et al. (2016) [47]	0	1	0	1	1	1	1	1	6	Some concerns
Golley et al. (2013) [48]	0	1	0	1	1	1	1	1	6	Some concerns
Jindal et al. (2021) [49]	0	1	1	1	1	1	1	1	7	Low risk
Lane et al. (2022) [50]	1	1	1	1	1	1	1	1	8	Low risk
Lee et al. (2017) [51]	0	1	0	1	1	1	1	1	6	Some concerns
Min et al. (2018) [52]	1	1	1	1	1	1	1	1	8	Low risk
Mozaffari-Khosravi et al. (2021) [34]	1	1	1	1	0	0	1	1	6	Some concerns
Negele et al. (2020) [24]	1	0	1	1	1	1	1	1	7	Low risk
Ozkan et al. (2020) [53]	1	0	1	1	0	0	1	1	5	Some concerns
Werneck et al. (2018) [54]	1	0	1	1	1	1	1	1	7	Low risk
Yaghtin et al. (2022) [55]	1	1	1	1	1	1	1	1	8	Low risk

Note: 0 = Criteria not met; 1 = Criteria met; Criteria for analysis: 1 = Were the inclusion criteria clearly defined in the sample? 2 = Were the study subjects and setting described in detail? 3 = Was the exposure measured validly and reliably? 4 = Were objective and standard criteria used to measure the condition? 5 = Were confounding factors identified? 6 = Were strategies mentioned to address confounding? 7 = Were the outcomes measured in a valid and reliable way? 8 = Was appropriate statistical analysis used?

**Table 2 nutrients-16-03345-t002:** Characteristics of the articles included in the review classified by type of results.

Type of Results	Reference	Country	Design	N	Age	Gender	Diet Measurements	PA Measurements	Sleep Measurements	Results
Diet	Beigrezaei et al. (2022) [45]	Iran	Cross-sectional	988	12–18 (14.52 ± 1.52)	F 988 (100%)	Eating Behavior Questionnaire	Modifiable Activity Questionnaire (MAQ) (MET-h/week)	Insomnia Severity Index (ISI)	Regular meal consumption → ↓ likelihood of insomnia Always eating breakfast → ↓ likelihood of insomnia Daily spicy food consumption → ↑ likelihood of insomnia
	Bhurosy and Thiagarajah (2020) [46]	United States	Cross-sectional	13,583	14–18	F 6052 (50.2%) M 6273 (49.8%)	Selected items from the YRBS survey	-	Elementos seleccionados de la encuesta YRBS	Higher daily intake of vegetables, milk and breakfast → ↑ likelihood of sleeping ≥8 h Higher daily soda intake → ↓ likelihood of sleeping ≥8 h
	Ferranti et al. (2016) [47]	Italy	Cross-sectional	1586	11–14 (12 ± 0.7)	F 716 (45.15%) M 870 (54.85%)	FFQ KIDMED	Cuestionario de Actividad Física para Adolescentes (PAQ-A)	Questionnaire on sleep patterns Pediatric Daytime Sleepiness Scale (PDSS)	Fruit and vegetable intake ↔ ↑ total and weekday sleep time Higher consumption of sweets, snacks and eating out ↔ ↓ sleep and directly related to PDSS Breakfast habit and physical activity ↔ no significant relation with sleep duration or PDSS KIDMED score ↔ direct linear relation with PDSS, weekday sleep time and total sleep time
	Golley et al. (2013) [48]	Australia	Cross-sectional	1516	12–16 (14.2 ± 1.3)	F 1114 (50.64%) M 1086 (49.36%)	Food consumption in the last 24 h	Multimedia Activity Recall for Children and Adults (MARCA)	Reported amount of time in the last 24 h	Late bedtime → poorer diet quality and ↑ intake of additional foods (energy-dense, nutrient-poor) Early bedtime → ↑ fruit and vegetable consumption Energy intake associated ↔ with sleep duration
	Lane et al. (2022) [50]	Iran	Cross-sectional	733	12–18 (14.51 ± 1.53)	F 733 (100%)	Daily, weekly, monthly and yearly consumption Quartiles of ultra-processed food intake (Q1, Q2, Q3 and Q4)	Modifiable Activity Questionnaire (MAQ) (MET-h/week)	Insomnia Severity Index (ISI)	↑ UP intake → ↑ likelihood of insomnia Participants in Q4 quartile of UP intake → ↑ likelihood of insomnia compared to Q1 (OR: 2.77 (insomnia) vs. Q1 (95% CI: 1.5–5.10, *p* < 0.01))
	Min et al. (2018) [52]	South Korea	Cross-sectional	118,462	12–18 (15 ± 1.7)	F 59,031 (49.83%) M 59,431 (50.17%)	Frequency of consumption of CERTAIN foods in the last seven days	-	Bedtime and wake-up time last seven days (week and weekend) Sleep quality (good, moderate, poor)	Longer sleep duration → associated with ↑ consumption of instant noodles, fruits, vegetables and milk Shorter sleep durations → associated with ↑ consumption of soda, fast food and sweets Poor sleep quality → ↓ intake of fruits, vegetables and milk Poor sleep quality → ↑ intake of soda, fast food, instant noodles and sweets
	Ozkan et al. (2020) [53]	Turkey	Cross-sectional	346	11–13 (11.9 ± 0.8)	F 170 (49.1%) M 176 (50.9%)	Face-to-face questionnaire Food consumption in the last 24 h	-	Face-to-face questionnaire Pittsburgh Sleep Quality Index (PSQI)	NO regular breakfast ↔ ↓ sleep quality vs. regular breakfast Skipping meals ↔ ↓ sleep quality vs. not skipping meals 3–5 h sleep ↔ ↓ sleep quality vs. 6–8 h and >8 h sleep (*p* < 0.05) ↓ saturated fatty acids ↔ good sleep quality (PSQI) (*p* = 0.040)
Diet	Yaghtin et al. (2022) [55]	Iran	Cross-sectional	733	12–18 (14.49 ± 1.5)	F 733 (100%)	Food Frequency Questionnaire (FFQ) of 147 foods mMED diet score	Modifiable Activity Questionnaire (MAQ) and converted to (MET)-hours/day	Insomnia Severity Index (ISI)	Inverse association between mMED diet score and level of insomnia
PA	Jindal et al. (2021) [49]	South Korea	Cross-sectional	59	12–18 (15.3 ± 1.9)	-	-	Actigrafía y acelerometro (7 días consecutivos sin clases)	Actigraphy and accelerometer (seven consecutive non-school days)	Adequate sleep → ↓ sedentary time and ↑ light PA time Early sleep schedule ↔ associated with more PA (independent of sleep duration)
	Negele et al. (2020) [24]	Germany	Cohort study	1223	15.59 ± 0.52	F 673 (55%) M 550 (45%)	-	Triaxial accelerometer/three categories (sedentary, lifestyle and MVPA)	Triaxial accelerometer Total sleep time and sleep quality	PA ↔ only improves sleep quality the following night Girls: Every 10 min. increase in MVPA ↔ ↑ SE Boys: ≥60 min MVPA/day ↔ ↓ SOL MVPA ↔ ↑ SE in both sexes; ↑ TST in girls, ↓ SOL in boys Increase in lifestyle activity ↔ ↑ SOL in both sexes
	Werneck et al. (2018) [54]	Brazil	Cross-sectional	100,839	11–18 (14.28)	F 52,280 (51.85%) M 48,559 (48.15%)	Self-administered questionnaires Frequency of candy and ultra-processed foods consumption in the last week	Self-administered questionnaires Modified IPAQ	Self-administered questionnaires Frequency of insomnia in the last 12 months	Higher levels of PA ↔ ↓ prevalence of insomnia in boys Moderate levels of PA ↔ ↓ insomnia, but high levels of PA (≥420 min/week) ↔ ↑ insomnia in girls
PA and diet	Lee et al. (2017) [51]	South Korea	Cross-sectional	31,407	15–18 (16.4 ± 0.94)	F 15,929 (50.7%) M 15,478 (49.3%)	Questionnaire on the frequency of consumption of CERTAIN foods in the last seven days	Fr. PA ≥ 60 min, fr. Vigorous PA and FM last seven days	Bedtime and wake-up time for the last five weekdays	Insufficient sleep (<7 h) → associated with risk factors Insufficient sleep (<7 h) → associated with ↓ likelihood of doing muscle-strengthening exercises Insufficient sleep (<7 h) → associated with ↑ likelihood of consuming salty crackers
	Mozaffari-Khosravi et al. (2021) [34]	Iran	Transversal	569	12–16 (14.24 ± 0.88)	-	Lifestyle habits questionnaire (nutrition and PA) Breakfast consumption, breakfast time, lunch, dinner, dinner time	Lifestyle habits questionnaire (nutrition and PA) PA ≥ 30 min. last seven days Hours in PE class per week	Pittsburgh Sleep Quality Index (PSQI)	Poor sleep quality ↔ associated with late dinner, late bedtime and low PA Significant relationship between good sleep quality and earlier bedtime Significant relationship between PA and sleep quality score Better sleep quality with moderate PA than with light PA

Note: ↑ = higher; ↓ = lower; ↔ = associated with; PA = physical activity; FFQ = Food Frequency Questionnaire; Fr. = frequency; UP = ultra-processed; SQ = sleep quality; SE = sleep efficiency; SOL = sleep onset latency; TAPH = time awake per hour after sleep onset; FM = muscle strength; MVPA = moderate to vigorous physical activity; F = female; M = male; IPAQ = International Physical Activity Questionnaire; YRBS = Youth Risk Behavior Surveillance System.

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
