# Peer review of "Association of Physical Activity and/or Diet with Sleep Quality and Duration in Adolescents: A Scoping Review"

_nutrients, 2024, doi:10.3390/nu16193345_

Round 1

Reviewer 1 Report (Previous Reviewer 2)

Comments and Suggestions for Authors

Dear Sirs,

I believe that with the changes made.

Author Response

Comments 1: Dear Sirs, I believe that with the changes made.

Author’s response (AR): Thank you for the review and comments. We would like to express our gratitude to Reviewer 1 for the time in reviewing our paper. We tried to improve the paper taking into consideration the recommendations that we received from the reviewers.

Reviewer 2 Report (New Reviewer)

Comments and Suggestions for Authors

I would like to express my deep gratitude for the opportunity to review your work. It is a privilege to contribute to the evaluation of such a detailed and well-structured study, which addresses highly relevant topics for adolescent well-being, such as physical activity, diet, and sleep. The quality of the research is evident in the methodology used and the clarity with which the results are presented.

Strengths of the study:

  • The study highlights the important combined role of physical activity and diet on sleep quality and duration in adolescents. This integrated approach is a significant contribution to the existing scientific literature.
  • The systematic review and narrative analysis are conducted in a rigorous and structured manner, providing a clear overview of the available evidence.
  • The use of the PECOS model for study inclusion is a methodological strength, ensuring the relevance and appropriateness of the included works.
  • The application of the JBI checklist for assessing the methodological quality of the included studies adds robustness to the analysis of the results.

Suggested modifications:

  • Lines 47-50: The statement that "11% to 30% of adolescents experience sleep disorders" could benefit from further exploration of the causes of this variability, including, if possible, a geographic or socio-economic analysis to better contextualize the data.

  • Lines 125-128: The description of the search equation could be enhanced by providing more detailed explanations of the use of descriptors and Boolean operators, to make the search process more replicable.

  • Lines 137-150: In defining the exclusion criteria, it might be helpful to clarify more precisely the rationale behind excluding studies conducted during the pandemic period. Including an explanatory note could prevent ambiguity about the applicability of this data.

  • Lines 202-204: I recommend providing more details on the studies that raise "some concerns" regarding methodology, clarifying the specific reasons why these studies were not completely excluded. This would contribute to greater transparency in the assessment of the risk of bias.

  • Lines 271-272: The statement that "sleep patterns and physical activity coexist and are closely correlated" could be strengthened by including specific examples of studies that confirm this link, along with statistical results to enhance the argument.

Parts to consider for elimination or revision:

  • Lines 158-160: The "snowball" strategy for identifying additional relevant studies might be redundant if it is not accompanied by a clear justification or example of how it enriched the number of included studies. If it did not produce significant results, it could be eliminated or scaled down.

Study limitations:

Although the study is well-structured, the main limitations that could affect the interpretation of the results are not explicitly indicated. Here are a few aspects to consider:

  • Generalizability: The fact that the study includes only articles in English or Spanish may limit the generalizability of the results globally. This should be discussed in the limitations.
  • Causality: The included studies are mostly observational, which prevents drawing causal conclusions between physical activity, diet, and sleep quality.
  • Self-reporting: Most measurements of physical activity, diet, and sleep are based on self-reported questionnaires, which can be subject to social desirability or memory biases. It is important to highlight this aspect and consider future use of more objective measures.

Author Response

Comments and Suggestions for Authors

I would like to express my deep gratitude for the opportunity to review your work. It is a privilege to contribute to the evaluation of such a detailed and well-structured study, which addresses highly relevant topics for adolescent well-being, such as physical activity, diet, and sleep. The quality of the research is evident in the methodology used and the clarity with which the results are presented.

Strengths of the study:

The study highlights the important combined role of physical activity and diet on sleep quality and duration in adolescents. This integrated approach is a significant contribution to the existing scientific literature.

The systematic review and narrative analysis are conducted in a rigorous and structured manner, providing a clear overview of the available evidence.

The use of the PECOS model for study inclusion is a methodological strength, ensuring the relevance and appropriateness of the included works.

The application of the JBI checklist for assessing the methodological quality of the included studies adds robustness to the analysis of the results.

Author’s response (AR): Thank you for the review and comments. First, we would like to express our gratitude to Reviewer 2 for taking the time to review our paper and for providing helpful comments/suggestions to improve the manuscript. We think that your observations have improved the manuscript. We have answered point-by-point in blue in this document and in red in the new version of the manuscript. We find your comments and recommendations positive and very constructive.

Suggested modifications:

Lines 47-50: The statement that "11% to 30% of adolescents experience sleep disorders" could benefit from further exploration of the causes of this variability, including, if possible, a geographic or socio-economic analysis to better contextualize the data.

AR: Thank you very much for the advice. Taking into consideration the reviewer's suggestion we have rewritten the sentence:

“Secondly, the situation is even more alarming among adolescents, with 11% to 30% experiencing sleep disorders. This variation in the prevalence of adolescents' sleep disorders could be due to age, region and culture. For example, older adolescents go to bed later and slept less [5]. In addition, Asian adolescents go to bed later than American adolescents and obtain less sleep than Europeans [6].”

  1. Gariepy, G.; Danna, S.; Gobiņa, I.; Rasmussen, M.; Gaspar de Matos, M.; Tynjälä, J.; Janssen, I. PhD.; Kalman, M. PhD.; Villeruša, A.; Husarova, D.; Brooks. F.; Elgar. F.J.; Klavina-Makrecka, S. MSc.; Šmigelskas, K.; Gaspar, T.; Schnohr, C. How Are Adolescents Sleeping? Adolescent Sleep Patterns and Sociodemographic Differences in 24 European and North American Countries. J Adolesc Health, 2020, 66(6S):S81-S88. doi: 10.1016/j.jadohealth.2020.03.013. PMID: 32446613.

Lines 125-128: The description of the search equation could be enhanced by providing more detailed explanations of the use of descriptors and Boolean operators, to make the search process more replicable.

AR: Thank you very much. We have modified the paragraph to make the search process more replicable:

A structured search was performed in four online databases (PubMed, Scopus, Web of Science, and ERIC) and concluded on March 20, 2023. The search terms included a combination of free text words for key concepts related to PA, diet, adolescents, and sleep. To enhance the search process, some Boolean operators (AND, OR, and NOT) with field identifiers or qualifiers (Title, Title/Abstract, All) were used. The following search equation was written in the search boxes of the mentioned databases to find relevant articles: (adolescents OR "young people" OR teenagers OR youth OR “high school” OR “secondary school”) AND ("physical activity" OR exercise OR sports OR fitness) AND ("dietary habits" OR "eating habits" OR "food consumption" OR nutrition) AND (sleep OR "sleeping habits" OR "bedtime routines" OR rest) AND ("sleep quality" OR "sleep disturbance" OR insomnia).

The detailed equation used in pubmed was the following: ("adolescences"[All Fields] OR "adolescency"[All Fields] OR "adolescent"[MeSH Terms] OR "adolescent"[All Fields] OR "adolescence"[All Fields] OR "adolescents"[All Fields] OR "adolescent s"[All Fields] OR "young people"[All Fields] OR ("adolescent"[MeSH Terms] OR "adolescent"[All Fields] OR "teenage"[All Fields] OR "teenager"[All Fields] OR "teenagers"[All Fields] OR "teenaged"[All Fields] OR "teenager s"[All Fields] OR "teenages"[All Fields]) OR ("adolescent"[MeSH Terms] OR "adolescent"[All Fields] OR "youth"[All Fields] OR "youths"[All Fields] OR "youth s"[All Fields]) OR "high school"[All Fields] OR "secondary school"[All Fields]) AND ("physical activity"[All Fields] OR ("exercise"[MeSH Terms] OR "exercise"[All Fields] OR "exercises"[All Fields] OR "exercise therapy"[MeSH Terms] OR ("exercise"[All Fields] AND "therapy"[All Fields]) OR "exercise therapy"[All Fields] OR "exercising"[All Fields] OR "exercise s"[All Fields] OR "exercised"[All Fields] OR "exerciser"[All Fields] OR "exercisers"[All Fields]) OR ("sport s"[All Fields] OR "sports"[MeSH Terms] OR "sports"[All Fields] OR "sport"[All Fields] OR "sporting"[All Fields]) OR ("fitness"[All Fields] OR "fitnesses"[All Fields])) AND ("dietary habits"[All Fields] OR "eating habits"[All Fields] OR "food consumption"[All Fields] OR ("nutrition s"[All Fields] OR "nutritional status"[MeSH Terms] OR ("nutritional"[All Fields] AND "status"[All Fields]) OR "nutritional status"[All Fields] OR "nutrition"[All Fields] OR "nutritional sciences"[MeSH Terms] OR ("nutritional"[All Fields] AND "sciences"[All Fields]) OR "nutritional sciences"[All Fields] OR "nutritional"[All Fields] OR "nutritionals"[All Fields] OR "nutritions"[All Fields] OR "nutritive"[All Fields])) AND ("sleep"[MeSH Terms] OR "sleep"[All Fields] OR "sleeping"[All Fields] OR "sleeps"[All Fields] OR "sleep s"[All Fields] OR "sleeping habits"[All Fields] OR "bedtime routines"[All Fields] OR ("rest"[MeSH Terms] OR "rest"[All Fields])) AND ("sleep quality"[All Fields] OR "sleep disturbance"[All Fields] OR ("insomnia s"[All Fields] OR "sleep initiation and maintenance disorders"[MeSH Terms] OR ("sleep"[All Fields] AND "initiation"[All Fields] AND "maintenance"[All Fields] AND "disorders"[All Fields]) OR "sleep initiation and maintenance disorders"[All Fields] OR "insomnia"[All Fields] OR "insomnias"[All Fields]))

However, this detailed equation lengthened the text too much, so we have not included it in the manuscript.

Lines 137-150: In defining the exclusion criteria, it might be helpful to clarify more precisely the rationale behind excluding studies conducted during the pandemic period. Including an explanatory note could prevent ambiguity about the applicability of this data.

AR: Thank you very much. We excluded studies investigating the variables during the lockdown period of the COVID-19 pandemic because the COVID-19 period was special period. In that period, the physical activity and sleep patterns changed (Ding et al., 201; Luong et al., 2024; Wilke et al., 2021). Following the reviewer's suggestion, we have included an explanatory sentence: “These studies were excluded because during this period the PA and sleep patterns changed considerably [39-41].”

In the references section we have included the mentioned references.

  1. Ding, K.; Yang, J.; Chin, M.K.; et al. Physical activity among adults residing in 11 countries during the COVID-19 pandemic lockdown. Int J Environ Res Public Health. 2021 18(13): 7056.
  2. Luong, N.; Mark, G.; Kulshrestha, J.; Aledavood, T. Sleep During the COVID-19 Pandemic: Longitudinal Observational Study Combining Multisensor Data With Questionnaires. JMIR Mhealth Uhealth. 2024, 3,12:e53389. https://doi.org/10.2196/53389. PMID: 39226100.
  3. Wilke, J.; Mohr, L.; Tenforde, A.S.; et al. A pandemic within the pandemic? Physical activity levels substantially decreased in countries affected by COVID-19. Int J Environ Res Public Health. 2021,18, 2235.

Lines 202-204: I recommend providing more details on the studies that raise "some concerns" regarding methodology, clarifying the specific reasons why these studies were not completely excluded. This would contribute to greater transparency in the assessment of the risk of bias.

AR: Thank you very much for the advice. Note that we changed from Unclear to Some concerns in Table 1 to maintain consistency with the text. In this scoping review, we have tried to consider all the research about the topic we studied (provided they meet the established inclusion criteria), regardless of the level of the research. This is why we did not exclude it from the paper.

However, in the 3.2 Quality assessment section, we have reflected this reality: “With scores of 7 and 8, seven studies demonstrate a low risk of bias [24,45,49,50,52,54,55], while six studies raise some concerns [34,46-48,51,53] with scores of 5 and 6.”

Lines 271-272: The statement that "sleep patterns and physical activity coexist and are closely correlated" could be strengthened by including specific examples of studies that confirm this link, along with statistical results to enhance the argument.

AR: Thank you very much. All of the studies mentioned in lines 265-272 show a correlation between PA and sleep. However, indeed, the written text may not have made it clear that these were the studies we were referring to. For this reason, we have rewritten the text: “Several studies found that higher levels of PA were associated with better sleep quality and duration [24,49,5,54]. According to Jindal et al., there is an association between less sedentary or lighter PA time and adequate sleep [49]. The study also found that more PA time was associated with an earlier bedtime, regardless of sleep duration. Similarly, Lee et al. observed a correlation between insufficient sleep (<7 hours) and several risk factors, specifically more sedentary time and a lower likelihood of engaging in muscle-strengthening exercises [51]. Negele et al. found that PA improves sleep quality on the night following PA practice [24]. Although the mentioned studies show that PA and sleep are correlated, the specific association of PA with sleep can be complex and may vary by sex.”

Parts to consider for elimination or revision:

Lines 158-160: The "snowball" strategy for identifying additional relevant studies might be redundant if it is not accompanied by a clear justification or example of how it enriched the number of included studies. If it did not produce significant results, it could be eliminated or scaled down.

AR: Thank you very much for the advice. We agree that the explanation of the snowball strategy needs to be clarified. The snowball strategy did not enrich more articles for the scoping review; nevertheless, although it did not report any results, we think that it is important to show that we did use this strategy. We have rewritten the sentence: "The search for articles was also carried out using the snowball strategy [39], whereby the reference sections of all relevant articles were examined. Nevertheless, this strategy did not enrich more articles for the scoping review.”

Study limitations:

Although the study is well-structured, the main limitations that could affect the interpretation of the results are not explicitly indicated. Here are a few aspects to consider:

Generalizability: The fact that the study includes only articles in English or Spanish may limit the generalizability of the results globally. This should be discussed in the limitations.

AR: Thank you very much for the advice, but we have already considered this limitation in lines 372-377.

Causality: The included studies are mostly observational, which prevents drawing causal conclusions between physical activity, diet, and sleep quality.

AR: Thank you very much for the advice, but we have already considered this limitation in lines 378-387.

Self-reporting: Most measurements of physical activity, diet, and sleep are based on self-reported questionnaires, which can be subject to social desirability or memory biases. It is important to highlight this aspect and consider the future use of more objective measures.

AR: Thank you very much for the advice. We had add this aspect in this way: “In addition, most measurements of PA, diet, and sleep are based on self-reported questionnaires. In the future, it would be important to consider the use of more objective measures in order to avoid social desirability or memory biases.”

Reviewer 3 Report (New Reviewer)

Comments and Suggestions for Authors

The idea of ​​this article is interesting and timely, my recommendations are the following;

Abstract - rewriting the last conclusion, possibly mentioning that it is about teenagers.

Lines 98-102 mentioned as the previous studies, but in the bibliographic indexes it is only 32, I recommend clarification by adding other bibliographic indexes or reformulating the sentence referring to a single study.

The review is well organized and all the specific aspects of this type of article are presented.

Author Response

Comments and Suggestions for Authors

The idea of ​​this article is interesting and timely, my recommendations are the following;

Author’s response (AR): Thank you for the review and comments. First, we would like to express our gratitude to Reviewer 3 for taking the time to review our paper and for providing helpful comments/suggestions to improve the manuscript. We think that your observations have improved the manuscript. We have answered point-by-point in blue in this document and in red in the new version of the manuscript. We find your comments and recommendations positive and very constructive.

Abstract - rewriting the last conclusion, possibly mentioning that it is about teenagers.

AR: Thank you very much for the advice. We have rewritten this sentence as follows: “In conclusion, the results underscore the importance of considering PA and diet as an appropriate approach to sleep quality and duration in adolescents. Studies analysing the interplay between PA, diet, and sleep in adolescents are scarce.”

Lines 98-102 mentioned as the previous studies, but in the bibliographic indexes it is only 32, I recommend clarification by adding other bibliographic indexes or reformulating the sentence referring to a single study.

AR: Thank you very much for the advice. We have rewritten the sentence as follows: "Furthermore, some studies have pointed out that PA and diet together can modify sleep patterns during adolescence [33,34].”

The review is well organised and all the specific aspects of this type of article are presented.

This manuscript is a resubmission of an earlier submission. The following is a list of the peer review reports and author responses from that submission.

Round 1

Reviewer 1 Report

Comments and Suggestions for Authors This review addresses the relationship between diet and/or physical activity (PA) and sleep in adolescents, a critical period marked by the emergence of sleep problems, making the scope of the paper relevant. However, I have significant reservations about the quality of the work presented. While the topic is undoubtedly interesting, extensive revisions are essential and should be considered by the authors.

Please see below comments for consideration:

A-    One of my major concerns is regarding the flow of ideas in the introduction which fails to articulate a distinct purpose for the research. Accordingly, the introduction section appears disjointed, resembling more of a scattered literature review than a cohesive foundation, thereby impeding the formulation of a concise and comprehensive research question. I advise the authors to revise the introduction taking into account the following points:

- The effect of PA and/or diet on sleep should be introduced from the very beginning without getting into the effect of these behaviors on health which are already well documented in the literature.
- The studies should be listed in order from the most general to the most specific.

- Factors related to PA (energy expenditure, intensity, timing) and diet (energy intake, macro and micro-nutrients) and their potential effect on sleep should be explained.

- More rational behind the potential interaction between physical activity and diet should be discussed in order to support the objective of this review.

B-     I advise the authors to clearly state the primary and secondary objectives (if existing) of this review. If the primary objective is to examine the combined influence of physical activity (PA) and diet on sleep (line 24-25: abstract ; line 100-101: introduction), why were studies focusing solely on PA and sleep or diet and sleep included (these objectives need to be added)?

C-    This systematic review scope is broad and a methodology of a scoping review looks more suitable. At least the authors should add more details on the exposure whether for physical activity or diet (quantitative, qualitative, and timing [i.e., is time features of eating pattern referred to as chrono-nutrition or timing of exercise been considered?] as well as sleep outcomes.

D-    Would you please explain why the following studies were not retained. The list is not exhaustive but in case of missing studies the authors should revise their systematic review.

- AdelantadoRenau, M., BeltranValls, M. R., EstebanCornejo, I., MartínezVizcaíno, V., SantaliestraPasías, A. M., & MolinerUrdiales, D. (2019). The influence of adherence to the Mediterranean diet on academic performance is mediated by sleep quality in adolescents. Acta Paediatrica, 108(2), 339-346.

- Ferranti, R., Marventano, S., Castellano, S., Giogianni, G., Nolfo, F., Rametta, S., ... & Mistretta, A. (2016). Sleep quality and duration is related with diet and obesity in young adolescent living in Sicily, Southern Italy. Sleep Science, 9(2), 117-122.

- Ievers-Landis, C. E., Kneifel, A., Giesel, J., Rahman, F., Narasimhan, S., Uli, N., & O’Riordan, M. (2016). Dietary intake and eating-related cognitions related to sleep among adolescents who are overweight or obese. Journal of pediatric psychology, 41(6), 670-679.

- Jansen, E. C., Baylin, A., Cantoral, A., Tellez Rojo, M. M., Burgess, H. J., O’Brien, L. M., ... & Peterson, K. E. (2020). Dietary patterns in relation to prospective sleep duration and timing among Mexico City adolescents. Nutrients, 12(8), 2305.

- Saidi, O., Rochette, E., Del Sordo, G., Doré, É., Merlin, É., Walrand, S., & Duché, P. (2021). Eucaloric balanced diet improved objective sleep in adolescents with obesity. Nutrients, 13(10), 3550.

E-     The discussion section appears as a restatement of the results section. This should be revised taking into account what the current review adds to the literature compared to previous reviews, such as:

- Alibabaei, Z., Jazayeri, S., Vafa, M., Feizy, Z., & Hezaveh, Z. S. (2021). The association between dietary patterns and quality and duration of sleep in children and adolescents: A systematic review. Clinical Nutrition ESPEN, 45, 102-110.

- Lang, C., Kalak, N., Brand, S., Holsboer-Trachsler, E., Pühse, U., & Gerber, M. (2016). The relationship between physical activity and sleep from mid adolescence to early adulthood. A systematic review of methodological approaches and meta-analysis. Sleep medicine reviews, 28, 32-45. Comments on the Quality of English Language

Improvements could be made

Author Response

This review addresses the relationship between diet and/or physical activity (PA) and sleep in adolescents, a critical period marked by the emergence of sleep problems, making the scope of the paper relevant. However, I have significant reservations about the quality of the work presented. While the topic is undoubtedly interesting, extensive revisions are essential and should be considered by the authors.

Author’s response (AR): Thank you for the review and comments. First of all, we would like to express our gratitude to Reviewer 1 for the time in reviewing our paper and for providing us helpful comments/suggestions to improve the manuscript. We think that your observations have improved the manuscript. We have answered point-by-point in blue in this document and in the new version of the manuscript. We find your comments and recommendations positive and very constructive.

A-    One of my major concerns is regarding the flow of ideas in the introduction which fails to articulate a distinct purpose for the research. Accordingly, the introduction section appears disjointed, resembling more of a scattered literature review than a cohesive foundation, thereby impeding the formulation of a concise and comprehensive research question. I advise the authors to revise the introduction taking into account the following points:

- The effect of PA and/or diet on sleep should be introduced from the very beginning without getting into the effect of these behaviors on health which are already well documented in the literature.

- The studies should be listed in order from the most general to the most specific.

- Factors related to PA (energy expenditure, intensity, timing) and diet (energy intake, macro and micro-nutrients) and their potential effect on sleep should be explained.

 - More rational behind the potential interaction between physical activity and diet should be discussed in order to support the objective of this review.

AR: Thank you very much for the advice. We have taken into consideration the reviewer's input and suggested changes. Even though the other reviewer told us that the introduction was appropriate, we have made changes in the introduction in an attempt to reflect the considerations. We hope you will be pleased those changes.

B-     I advise the authors to clearly state the primary and secondary objectives (if existing) of this review. If the primary objective is to examine the combined influence of physical activity (PA) and diet on sleep (line 24-25: abstract; line 100-101: introduction), why were studies focusing solely on PA and sleep or diet and sleep included (these objectives need to be added)?
AR: Thank you very much. Taking into account this suggestion and the suggestions of the reviewer 2 we have rewritten the objectives like this: “Therefore, the main objective of this scoping review was to assess the extant research on investigating association of PA and diet on the quality and duration of sleep in adolescents. Secondary objectives were to assess the evidence on the association of PA on the quality and duration of sleep in adolescents and on the association of diet on the quality and duration of sleep in adolescents”.

C-    This systematic review scope is broad and a methodology of a scoping review looks more suitable. At least the authors should add more details on the exposure whether for physical activity or diet (quantitative, qualitative, and timing [i.e., is time features of eating pattern referred to as chrono-nutrition or timing of exercise been considered?] as well as sleep outcomes.
AR: Thank you very much. After analyzing the suggestions made by both reviewers, we have decided to change the study from a systematic review to a scoping review. Unfortunately, the outcomes mentioned have not been taken into account in this study in the results section. However, we consider that these aspects are addressed in the discussion section.
D-    Would you please explain why the following studies were not retained. The list is not exhaustive but in case of missing studies the authors should revise their systematic review.
AR: Thank you very much. We appreciate the suggestion and we have analyzed the studies and following we detail the reasons for their non-inclusion in the present research.

- Adelantado‐Renau, M., Beltran‐Valls, M. R., Esteban‐Cornejo, I., Martínez‐Vizcaíno, V., Santaliestra‐Pasías, A. M., & Moliner‐Urdiales, D. (2019). The influence of adherence to the Mediterranean diet on academic performance is mediated by sleep quality in adolescents. Acta Paediatrica, 108(2), 339-346.

The present study was not taken into consideration as it does not focus on the research topic. The present study examined the association between diet and academic performance and its effect on sleep, so we thought that is not suitable for our study.

- Ferranti, R., Marventano, S., Castellano, S., Giogianni, G., Nolfo, F., Rametta, S., ... & Mistretta, A. (2016). Sleep quality and duration is related with diet and obesity in young adolescent living in Sicily, Southern Italy. Sleep Science, 9(2), 117-122.
The present study was already taken into consideration and analyzed in the scoping review. It was included in Table 1.

- Levers-Landis, C. E., Kneifel, A., Giesel, J., Rahman, F., Narasimhan, S., Uli, N., & O’Riordan, M. (2016). Dietary intake and eating-related cognitions related to sleep among adolescents who are overweight or obese. Journal of pediatric psychology, 41(6), 670-679.

The reason for excluding this study was that specific study samples with overweight or obese can bias the correlation coefficient (r) between diet, physical activity and sleep, since, if the range of values in the variables is narrower than in the general population, r tends to be low and vice versa (Rong et al., 2000). We have specified this aspect in the method section. The authors took this aspect for granted, but, as the reviewer suggests, it is preferable to specify it.

Rong Y. Statistical methods and pitfalls in environmental data analysis. Environ Forensics 2000; 1: 213–220

 - Jansen, E. C., Baylin, A., Cantoral, A., Tellez Rojo, M. M., Burgess, H. J., O’Brien, L. M., ... & Peterson, K. E. (2020). Dietary patterns in relation to prospective sleep duration and timing among Mexico City adolescents. Nutrients, 12(8), 2305.
Subjects aged between 9 and 17 years participated in the study by Jansen et al (2020). Although the study showed sleep results stratified by age, it did not show the relationship between sleep and diet stratified by age. Therefore, this study was excluded because its population was outside the 11 to 18 age range and did not disaggregate data for participants within this age range.

 - Saidi, O., Rochette, E., Del Sordo, G., Doré, É., Merlin, É., Walrand, S., & Duché, P. (2021). Eucaloric balanced diet improved objective sleep in adolescents with obesity. Nutrients, 13(10), 3550.

The present study is an experimental work and therefore meets exclusion criterion 8 (studies that conducted an intervention). Likewise, in the study we assess the evidence that analyze the relationship between the different parameters studied and not the influence, and Saidi et al. analyze the influence, the effect, so it has not been taken into account.

E-     The discussion section appears as a restatement of the results section. This should be revised taking into account what the current review adds to the literature compared to previous reviews, such as:

- Alibabaei, Z., Jazayeri, S., Vafa, M., Feizy, Z., & Hezaveh, Z. S. (2021). The association between dietary patterns and quality and duration of sleep in children and adolescents: A systematic review. Clinical Nutrition ESPEN, 45, 102-110.

AR: Thank you very much for the comment. This study was taken into account in the introduction, but afterwards, we have not taken into account in the discussion. We have taken into consideration this study in the discussion of the new version.

- Lang, C., Kalak, N., Brand, S., Holsboer-Trachsler, E., Pühse, U., & Gerber, M. (2016). The relationship between physical activity and sleep from mid adolescence to early adulthood. A systematic review of methodological approaches and meta-analysis. Sleep medicine reviews, 28, 32-45.

AR: Thank you very much for the suggestion. We have added this study in the discussion. We believe that the discussion is now stronger as a result.

Comments on the Quality of English Language

Improvements could be made

AR: Thank you very much. The manuscript has been sent to proof reading service and improvements have been made in the English language.

Reviewer 2 Report

Comments and Suggestions for Authors

Dear All,

I find the research carried out interesting. I have some comments that are aimed at improving what you already have. I hope to be helpful.

The introduction seems appropriate to me, since it provides background information on what the scientific literature on the topic of study states. I was only left with doubt regarding the research problem, and why it is necessary to carry out the investigation. In line 94 they point out that there are gaps in the literature, I would like to know what those gaps are, and why it is necessary to study them (what happens if those gaps are not studied?).

What also leaves me doubtful is the objective, since it talks about relationship or influence. Doubts that become greater when reading their results, since they only talk about relationships.

Furthermore, in the summary they describe that the objective is to determine effect and influence.

I think it is necessary to analyze and just leave the appropriate concept.

The methodology chapter seems correctly described to me. Just like the results.

When analyzing the results described and the discussion raised, I believe that their work is more of a scoping review than a systematic review. I say this because the results generally describe what was found. And especially when reading the discussion chapter, since there are not many comparisons made with related research, or because, for example, what was found in the quality evaluation of the articles, and its possible relationship with the results obtained, is not discussed. they report.

I think that if you want to continue with a systematic review, you should do more work on what is described, or if you prefer you should switch to a scoping review. Analyze and decide.

I hope I have been helpful.

Greetings.

Author Response

I find the research carried out interesting. I have some comments that are aimed at improving what you already have. I hope to be helpful.

Author’s response (AR): Thank you for the review and positive comments. First of all, we would like to express our gratitude to Reviewer 2 for the time in reviewing our paper and for providing us helpful comments/suggestions to improve the manuscript. We are very proud of your comments. We think that your observations have improved the manuscript. We have answered point-by-point in blue in this document and in the new version of the manuscript. We find your comments and recommendations positive and very constructive.

The introduction seems appropriate to me, since it provides background information on what the scientific literature on the topic of study states. I was only left with doubt regarding the research problem, and why it is necessary to carry out the investigation. In line 94 they point out that there are gaps in the literature, I would like to know what those gaps are, and why it is necessary to study them (what happens if those gaps are not studied?).

AR: Thank you very much for the comment. The gap that we wrote in line 94 is that previous work had analyzed the association of physical activity on sleep or the association of diet on sleep but not the association of both parameters on sleep, as shown in the following lines (lines 95-99): “while systematic reviews have so far been conducted to explore the influence of PA on sleep [20,21,23,30] or the relationship between diet and sleep [24,31], no review has been found to comprehensively address the association of both variables on sleep in adolescents, that is, the effect of PA and diet together on sleep quality and quantity”.

In addition, in relation to the comments made by reviewer 1, some substantial changes have been made in the introductory section.

What also leaves me doubtful is the objective, since it talks about relationship or influence. Doubts that become greater when reading their results, since they only talk about relationships.

Furthermore, in the summary they describe that the objective is to determine effect and influence. I think it is necessary to analyze and just leave the appropriate concept.

AR: Thank you very much. We reviewed articles that analyzed the association, so we have rewritten the influence like association. We have rewritten the objectives like this: Therefore, the main objective of this scoping review was to assess the extant research on investigating association of PA and diet on the quality and duration of sleep in adolescents. Secondary objectives were to assess the evidence on the association of PA on the quality and duration of sleep in adolescents and on the association of diet on the quality and duration of sleep in adolescents.”.

The methodology chapter seems correctly described to me. Just like the results.

AR: Thank you very much for the positive comment. 

When analyzing the results described and the discussion raised, I believe that their work is more of a scoping review than a systematic review. I say this because the results generally describe what was found. And especially when reading the discussion chapter, since there are not many comparisons made with related research, or because, for example, what was found in the quality evaluation of the articles, and its possible relationship with the results obtained, is not discussed. they report.

I think that if you want to continue with a systematic review, you should do more work on what is described, or if you prefer you should switch to a scoping review. Analyze and decide.

AR: Thank you very much. After analyzing the suggestions made by both reviewers, we have decided to change the study from a systematic review to a scoping review.

Round 2

Reviewer 1 Report

Comments and Suggestions for Authors

I have no further comments.

Author Response

Comments 1: I have no further comments.

Response 1: Thanks for your comments in the previous review which has helped to improve the manuscript.

Reviewer 2 Report

Comments and Suggestions for Authors

Dear,

your work has improved.

Author Response

Comments 1: your work has improved..

Response 1: Thanks for your comments in the previous review which has helped to improve the manuscript.